# Ultrafast structural rearrangement dynamics induced by the photodetachment of phenoxide in aqueous solution

Tushar Debnath [1], Muhammad Shafiq Bin Mohd Yusof[1], Pei Jiang Low[1] & Zhi-Heng Loh[1,2]

The elementary processes that accompany the interaction of ionizing radiation with biologically relevant molecules are of fundamental importance. However, the ultrafast structural rearrangement dynamics induced by the ionization of biomolecules in aqueous solution remain hitherto unknown. Here, we employ femtosecond optical pump-probe spectroscopy to elucidate the vibrational wave packet dynamics that follow the photodetachment of phenoxide, a structural mimic of tyrosine, in aqueous solution. Photodetachment of phenoxide leads to wave packet dynamics of the phenoxyl radical along 12 different vibrational modes. Eight of the modes are totally symmetric and support structural rearrangement upon electron ejection. Comparison to a previous photodetachment study of phenoxide in the gas phase reveals the important role played by the solvent environment in driving ultrafast structural reorganization induced by ionizing radiation. This work provides insight into the ultrafast molecular dynamics that follow the interaction of ionizing radiation with molecules in aqueous solution.

---

[1] Division of Chemistry and Biological Chemistry, School of Physical and Mathematical Sciences, Nanyang Technological University, 21 Nanyang Link, Singapore 637371, Singapore. [2] Centre for Optical Fibre Technology, The Photonics Institute, Nanyang Technological University, Singapore 639798, Singapore. Correspondence and requests for materials should be addressed to Z.-H.L. (email: zhiheng@ntu.edu.sg)

The ionization of biomolecules in aqueous solution yields radical species and hydrated electrons, which in turn trigger a cascade of chemical reactions that form the basis of radiation chemistry and radiation biology. While many studies have elucidated the ensuing solvation dynamics of the electron injected into the solvent by ionization, the ultrafast vibrational dynamics of the molecular radical species remains largely unknown. Ultrafast structural rearrangement impacts the chemical reactivities, redox potentials, and electron transfer rates involving radical species[1–5], which in turn influence important biological processes such as long-range electron transfer in redox-active proteins[6–8] and charge transport in DNA[9–11]. In the absence of time-domain measurements, analysis of X-ray emission spectra, Auger electron spectra, and ab initio simulations reveal ultrafast X-ray-induced proton transfer dynamics in liquid water[12–15] and small biomolecules in aqueous solutions[16] occurring within the 4-fs lifetime of the oxygen 1 s core level lifetime. In addition, intermolecular Coulombic decay leading to Coulomb explosion has also been observed in coincidence spectroscopy and ab initio studies of hydrated small molecules in the gas phase[17–20].

In general, photoionization or photodetachment leads to ultrafast structural rearrangement due to the different equilibrium molecular geometries before and after the removal of an electron. In the case of gas-phase species, vibrational progressions observed in photoelectron spectra encode the participation of the various vibrational modes in the structural reorganization[21]. In the condensed phase, such an approach becomes conceivably challenging due to the general absence of vibrationally resolved features in the inhomogeneously broadened photoelectron spectra of solution species[22,23]. Indeed, the ~ 1-eV-broad photoelectron spectral features would obfuscate the typical ~ 0.01–0.1-eV vibrational level spacings. As an alternative, we propose to employ optical pump-probe spectroscopy with few-cycle laser pulses to observe vibrational wave packet motion launched by impulsive ionization or detachment of solutes in aqueous solution. In the present study, we demonstrate the feasibility of this approach on the phenoxide ion (PhO⁻; Fig. 1) in aqueous solution. Impulsive photodetachment gives rise to vibrational wave packet motion along normal modes with a non-vanishing displacement between the equilibrium geometries of the PhO⁻ anion and the phenoxyl

radical (PhO·; Fig. 2). In spectrally-resolved pump-probe spectroscopy, vibrational wave packet motion manifests itself as energy- and amplitude modulations of the differential absorption signal as a function of pump-probe time delay. The time-domain signal can be Fourier transformed to yield a set of vibrational frequencies. Akin to the origin of Franck-Condon-active modes in photoelectron spectroscopy, the vibrational frequencies that appear in the pump-probe signal encode the identity of normal modes along which structural rearrangement occurs upon photodetachment.

There are several considerations for the choice of phenoxide as the target of our present study. First, the salt form used in the experiments, sodium phenoxide, is highly soluble in water, thereby allowing us to perform experiments with samples of sufficiently high concentration (~ 0.2 M). Second, phenoxide has a considerably lower vertical ionization potential (7.1 eV)[23] than liquid water (11.16 eV)[22], hence strongly favoring the strong-field multiphoton ionization of phenoxide over the solvent. Third, the absorption signature of the phenoxyl radical due to the $^2B_1 \rightarrow$ $^2B_1$ transition ($\lambda_{max} = 392$ nm, $\varepsilon_{max} = 3.2 \times 10^3$ M$^{-1}$cm$^{-1}$)[24,25] is shifted away from the strong and broad absorption band of the hydrated electron by-product ($\lambda_{max} = 720$ nm, $\varepsilon_{max} = 2.27 \times 10^4$ M$^{-1}$cm$^{-1}$)[26], therefore facilitating the spectroscopic observation of the PhO· radical itself. Fourth, the availability of the high-resolution photodetachment spectrum[27] of phenoxide enables comparison between the gas-phase measurement results and those obtained from our study of the aqueous form. Finally, the conjugate acid of phenoxide, phenol (PhOH), is the chromophore in the redox-active amino acid tyrosine, which in turn suggests that the phenoxyl radical can serve as a model for the tyrosyl radical, a key intermediate in numerous biochemical redox[8,28] and proton-coupled electron transfer reactions[29–31].

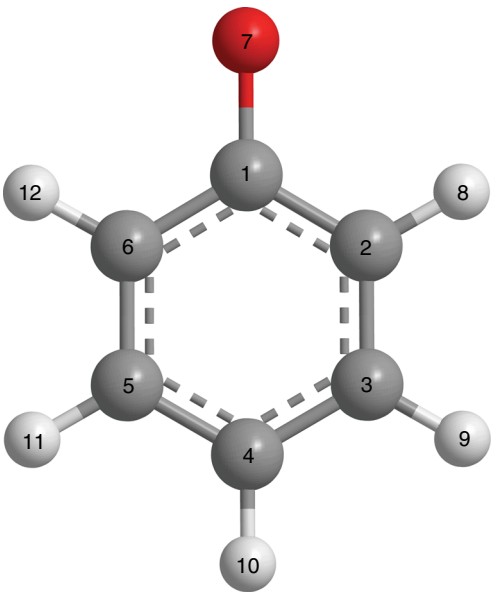

**Fig. 1** Structure of the phenoxide molecule. The carbon atoms are in gray, the hydrogen atoms are in white, and the oxygen atom is in red

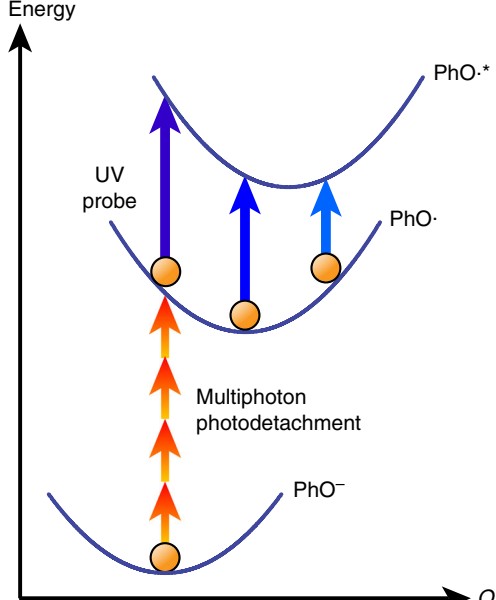

**Fig. 2** Vibrational wave packet generation by multiphoton photodetachment. Photodetachment of aqueous phenoxide by a strong-field, few-cycle laser pulse (red arrows) launches vibrational wave packet motion in the phenoxyl radical, which is subsequently probed by its time-dependent UV absorption (blue arrows). The potential energy curves of the initial phenoxide electronic ground state (PhO⁻), phenoxyl radical ground state produced by photodetachment (PhO•), and the phenoxyl radical excited state accessed by the UV probe pulse (PhO•*) are shown. Wave packet motion involves normal modes $Q_i$ whose minima are displaced relative to those of the phenoxide ion

In this work, photodetachment of phenoxide occurs in the strong-field regime and is effected by intense, few-cycle visible laser pulses. After a variable time delay, a near-UV pulse of 10-fs duration records the differential absorption ($\Delta A$) due to the formation of the phenoxyl radical product. Pronounced modulations in the differential absorption spectrum are observed as a function of time delay. Fourier transform of these time traces yields 12 vibrational frequencies of the phenoxyl radical, whose assignments reveal the identities of vibrational modes that participate in ultrafast photodetachment-induced structural rearrangement. The observation of a multitude of vibrational modes coupled to the detachment transition should be contrasted with the Franck-Condon progression observed only for a single vibrational mode in the gas-phase photodetachment study. Such disparate behavior points to the importance of the solvent environment in determining the extent of structural rearrangement that accompanies photodetachment. Our results provide the first glimpse into the ultrafast vibrational dynamics that accompany the photoionization or photodetachment of molecules of biological relevance in aqueous solution.

## Results

**Time-resolved differential absorption spectra**. The differential absorption spectra collected as a function of time delay is shown in Fig. 3a. Strong-field multiphoton photodetachment of phenoxide yields the phenoxyl radical and the accompanying hydrated electron. The former appears as a prominent absorption band at 400 nm alongside the vibronic feature at 382 nm, in good agreement with the previously reported absorption spectrum of PhO· in aqueous solution, whereas the latter gives rise to the increasing

$\Delta A$ signal that appears towards the long-wavelength region of the probe spectral range. Previous studies have shown that 4.7-eV photoexcitation of phenoxide to the $S_1$ ($\pi\pi^*$) state yields the phenoxyl radical via delayed electron ejection[24,32]. In our work, we rule out the formation of PhO· via the dense manifold of electronic excited states[33] of PhO⁻ because the pump-power dependence measurement shows that photodetachment occurs directly via a four-photon process (see Supplementary Fig. 1 and Supplementary Note 1). Moreover, contributions from excited-state dynamics in the presence of an intense laser field are expected to be negligible because these states are located in close energy proximity to the photodetachment threshold and therefore under rapid strong-field photodetachment on the sub-cycle timescale[34,35]. Further evidence for the immediate production of the phenoxyl radical via strong-field photodetachment comes from examining the $\Delta A$ time traces (Fig. 3b), which show an offset immediately following time-zero. This offset, absent in the time traces collected for ionized pure liquid water under experimental conditions identical to those employed for aqueous phenoxide (see Supplementary Fig. 2), is consistent with the instantaneous formation of the phenoxyl radical by strong-field photodetachment. Note that the rising $\Delta A$ signal that is observed atop the offset originates from hydrated electron formation[36,37] instead of the delayed formation of the phenoxyl radical via the $S_1$ state of phenoxide; a similar rise is observed in the control experiments performed on pure liquid water (see Supplementary Fig. 2). The precursor to the hydrated electron is an electron that is initially injected into the conduction band of liquid water by ionization. Due to its spatial delocalization, the formation of a contact pair between the conduction-band electron and the phenoxyl radical is not expected to occur. As such, the observed ultrafast dynamics is unlikely to originate from such a contact pair.

**Photodetachment-induced vibrational wave packet dynamics**. The differential absorption spectrum reveals pronounced modulations as a function of time delay, evident as well in the $\Delta A$ signal recorded at probe wavelengths of 400 nm and 406 nm (Fig. 3b). These modulations encode vibrational wave packet motion of the PhO· radical. Impulsive stimulated Raman pumping of phenoxide can be excluded as the origin of the observed vibrational wave packet dynamics because this differential absorption band and its time-domain oscillations disappear when the peak intensity of the pump pulse is attenuated to suppress strong-field photodetachment. Moreover, the absorption spectrum of phenoxide does not overlap with the probe spectral range (see Supplementary Fig. 5).

Fast Fourier transform (FFT) of the differential absorption spectra recorded as a function of time delay yields the 2D FFT power spectrum (Fig. 4a). The 2D FFT power spectrum reveals the distribution of oscillation frequencies over a range of probe wavelengths. Figure 4b shows the FFT power spectrum for the time trace collected at 406 nm, where the FFT power of the 528-cm⁻¹ oscillation is a maximum. Aside from the most prominent feature at 528 cm⁻¹, it is evident that a multitude of vibrational modes are coherently excited upon photodetachment. Table 1 summarizes the observed vibrational frequencies, along with their normalized FFT powers. These vibrational modes are also evident from the first-moment time trace (Fig. 5a), given as

$$\left\langle E^{(1)}(t) \right\rangle = \frac{\int_{E_i}^{E_f} dE\, E\, \Delta A(E,t)}{\int_{E_i}^{E_f} dE\, \Delta A(E,t)}, \quad (1)$$

where $\Delta A(E,t)$ is the differential absorption signal at probe photon energy $E$ and time delay $t$, and $E_i$ and $E_f$ define the range

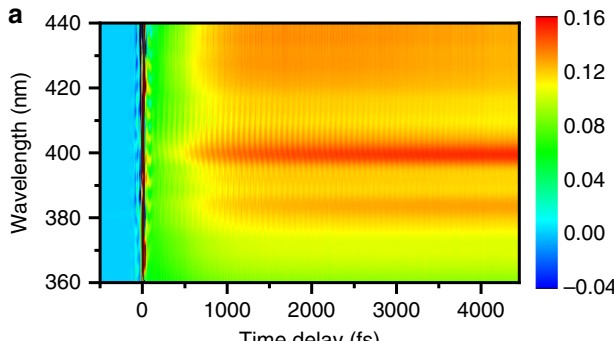

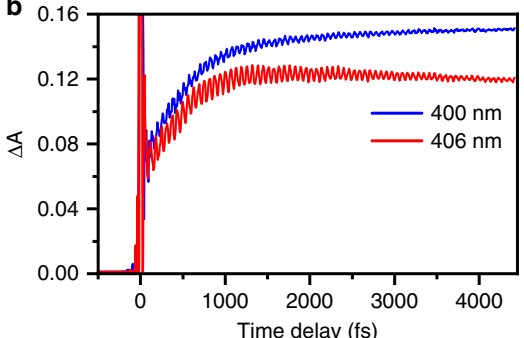

**Fig. 3** Time-resolved differential absorption spectra. **a** Differential absorption spectra of photodetached phenoxide in aqueous solution as a function of pump-probe time delay. The color scale gives the $\Delta A$ value. **b** Time-traces at 400 nm (blue) and 406 nm (red), showing pronounced modulations due to vibrational wave packet dynamics. Source data are provided as a Source Data file

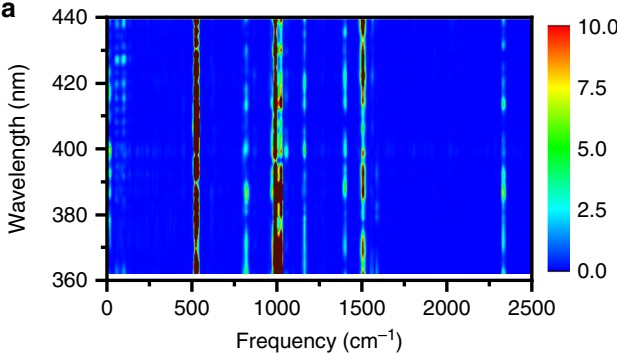

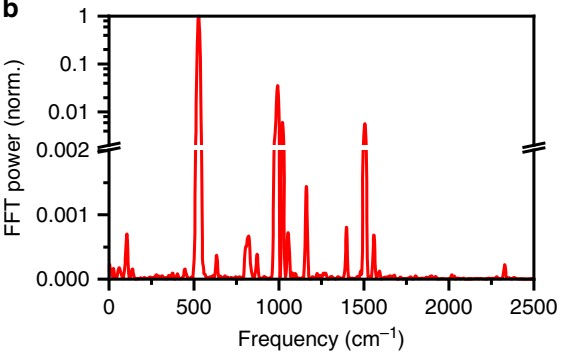

**Fig. 4** Fast Fourier transform (FFT) power spectra of photodetached aqueous phenoxide. **a** 2D FFT power spectrum of photodetached phenoxide in aqueous solution computed over the entire probe spectral range. The color scale gives the FFT power. **b** Line-out of normalized FFT power spectrum at 406 nm. Note the linear and logarithmic scales before and after the break in the vertical axis. Source data are provided as a Source Data file

over which the first-moment is computed; here, $E_i$ and $E_f$ correspond to 2.99 and 3.18 eV, respectively. Oscillatory features in $\langle E^{(1)}(t) \rangle$ originate from the modulation of the probe transition energy as vibrational wave packet motion occurs between the inner and outer turning points of the PhO· radical potential[38]. The FFT power spectrum of $\langle E^{(1)}(t) \rangle$ reveals the vibrational modes that drive such displacive wave packet motion (Fig. 5b). Time-domain analysis of the first-moment time trace (Fig. 5a) suggests that the 528-cm$^{-1}$ oscillation has a dephasing time of $1.86 \pm 0.07$ ps. All the other observed vibrational modes have dephasing times exceeding 0.5 ps. Similarly long vibrational dephasing times have been observed for other molecules in the condensed phase[39,40]. Such long dephasing times are suggestive of slow intramolecular vibrational relaxation (IVR) and a negligible effect of the solvent rearrangement on the vibrational frequencies of the phenoxyl radical.

**Normal mode assignment and origin of vibrational coherences.** The vibrational frequencies of the phenoxyl radical have been obtained from infrared spectroscopy of photochemically generated PhO· in a frozen argon matrix[41] as well as from numerous ab initio calculations[17,18,42,43] of the isolated PhO· radical. To assign the vibrational frequencies observed for the aqueous phenoxyl radical, we performed density functional theory (DFT) calculations on a microhydrated phenoxyl radical embedded in a polarizable continuum. Table 1 lists the calculated vibrational frequencies and their assignments. The experimentally observed vibrational frequencies are in good agreement with the calculated frequencies. While most of the observed modes have $A_1$

symmetry, four of the modes are non-totally symmetric. In the absence of a calculated vibrational mode in the vicinity of 1054 cm$^{-1}$, we assign it to the first overtone of the most intense peak at 528 cm$^{-1}$ (see below). The C–H stretching modes at >3000 cm$^{-1}$ are not discernible, most likely due to the limited 12-fs time resolution of the experimental setup. Note that impulsive stimulated Raman pumping of $N_2$ and $O_2$, originating from the measurements being performed in air, yields signatures of their vibrational and rotational wave packets in the FFT power spectrum. The frequency components 1555 cm$^{-1}$ and 2330 cm$^{-1}$ correspond to the $O_2$ and $N_2$ stretching frequencies[44], whereas the band of low frequency components (<160 cm$^{-1}$) are dominated by rotational coherences[45] (see Supplementary Fig. 6 and Supplementary Note 2).

It is important to note that a given vibrational mode would appear in the first-moment time trace $\langle E^{(1)}(t) \rangle$ only if a displacement exists between the potential energy curves for PhO$^-$ and PhO· along that mode. The displaced minima in turn requires the vibrational mode to be totally symmetric, or equivalently, Franck-Condon active. Aside from the mode at 1054 cm$^{-1}$ and the vibrational modes of $O_2$ (1559 cm$^{-1}$) and $N_2$ (2330 cm$^{-1}$), the remaining seven frequencies can all be attributed to modes of $A_1$ symmetry. The contribution of the 1054-cm$^{-1}$ mode to $\langle E^{(1)}(t) \rangle$ strongly supports its assignment to the first overtone of the dominant mode of $A_1$ symmetry at 528 cm$^{-1}$. This result suggests that anharmonicity of the 528-cm$^{-1}$ mode is very small, consistent with the first overtone of the same mode appearing at exactly twice the fundamental frequency in the gas-phase photodetachment study (519 cm$^{-1}$ fundamental vs. 1038 cm$^{-1}$ first overtone)[27]. The small anharmonicity suppresses coupling between different vibrational modes, resulting in slow IVR and is therefore consistent with the observed long dephasing times of the vibrational wave packets.

## Discussion

The vibrational wave packet dynamics launched by photoionization or photodetachment share a common origin with those that accompany the photoexcitation of molecules in the condensed phase[46]. The totally symmetric, Franck-Condon-active modes arise from displacement between the PhO$^-$ and PhO· potential energy surfaces along those modes, whereas non-totally symmetric modes involve non-displaced potential energy surfaces with different curvatures. The former yields vibrational wave packet motion between the inner and outer turning points of the PhO· potential energy curves, which in turn yields a time-dependent first-moment $\langle E^{(1)}(t) \rangle$. On the other hand, the latter produces a breathing-type wave packet, whose time-dependent width but fixed center of gravity results in a pure oscillation of the spectral amplitude as a function of time delay without any accompanying modulation of the first-moment.

To elucidate the effect of the solvent environment on the photodetachment-induced structural reorganization, we compare our experimental results to the high-resolution photodetachment spectrum of phenoxide measured in the gas phase[27]. The photodetachment spectrum reveals a Franck-Condon progression with peaks spaced apart by 519 cm$^{-1}$, which corresponds to the frequency of the C–C–C bending mode. (Note that additional peaks that appear in the gas-phase photodetachment spectrum of phenoxide could be due to vibrational progressions involving other modes, although the origin of these peaks was not discussed in ref. 27; see also Supplementary Note 3.) The same mode dominates the vibrational wave packet dynamics observed in aqueous solution, albeit shifted to a frequency of 528 cm$^{-1}$. However, the observation of only a single, assigned vibrational mode in the gas

**Table 1 Experimental and calculated vibrational frequencies of the phenoxyl radical in aqueous solution**

| | Freq. (cm⁻¹) | Power (norm.) | Calcd. freq. (cm⁻¹) | Assignment | Symmetry |
|---|---|---|---|---|---|
| 1 | | | 199 | Boat def / CO wag | $B_1$ |
| 2 | | | 376 | Ring def | $A_2$ |
| 3 | | | 437 | CO bend | $B_2$ |
| 4 | | | 473 | Boat def / CO wag | $B_1$ |
| 5 | 528 | 1 | 541 | CCC bend | $A_1$ |
| 6 | | | 591 | CCC in-plane bend | $B_2$ |
| 7 | 633 | $3.7 \times 10^{-4}$ | 643 | Chair def / CH wag | $B_1$ |
| 8 | | | 793 | CH wag | $A_2$ |
| 9 | 804 | $5.0 \times 10^{-4}$ | 809 | Chair def / CO CH wag | $B_1$ |
| 10 | 822 | $6.7 \times 10^{-4}$ | 818 | Ring breath / CCC bend | $A_1$ |
| 11 | 871 | $3.9 \times 10^{-4}$ | 937 | CH wag / boat def | $B_1$ |
| 12 | 991 | $3.5 \times 10^{-2}$ | 988 | CCC trig bend | $A_1$ |
| 13 | 972 | $2.3 \times 10^{-3}$ | 999 | HCCH torsion | $A_2$ |
| 14 | 1022 | $6.1 \times 10^{-3}$ | 1006 | Ring breath / CH bend | $A_1$ |
| 15 | | | 1007 | HCCH torsion | $B_1$ |
| | 1054 | $7.3 \times 10^{-4}$ | | CCC bend first overtone | $A_1$ |
| 16 | | | 1099 | CH bend / CC str | $B_2$ |
| 17 | 1162 | $1.5 \times 10^{-3}$ | 1164 | CH bend | $A_1$ |
| 18 | | | 1180 | CH bend / CC str | $B_2$ |
| 19 | | | 1295 | CC str / CH bend | $B_2$ |
| 20 | | | 1354 | CC str / CH bend | $B_2$ |
| 21 | 1397 | $8.0 \times 10^{-4}$ | 1416 | CH bend / CC str | $A_1$ |
| 22 | | | 1437 | CH bend / CC str | $B_2$ |
| 23 | 1506 | $5.7 \times 10^{-3}$ | 1511 | CO str | $A_1$ |
| 24 | | | 1542 | CC str / CH bend | $B_2$ |
| 25 | | | 1597 | CC str | $A_1$ |
| 26 | | | 3185 | CH str | $A_1$ |
| 27 | | | 3189 | CH str | $B_2$ |
| 28 | | | 3198 | CH str | $A_1$ |
| 29 | | | 3203 | CH str | $B_2$ |
| 30 | | | 3208 | CH str | $A_1$ |
| | 1555 | $6.9 \times 10^{-4}$ | | | $O_2$ |
| | 2330 | $2.3 \times 10^{-4}$ | | | $N_2$ |

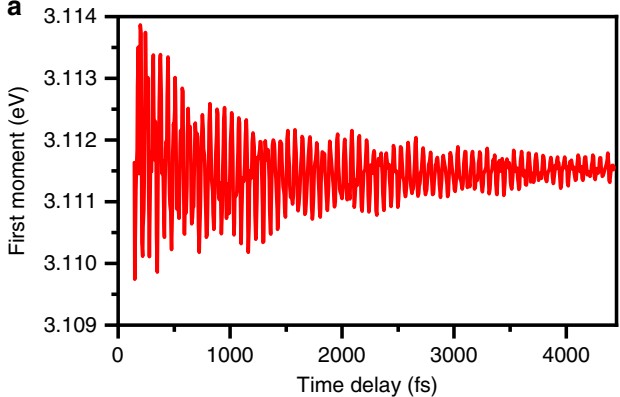

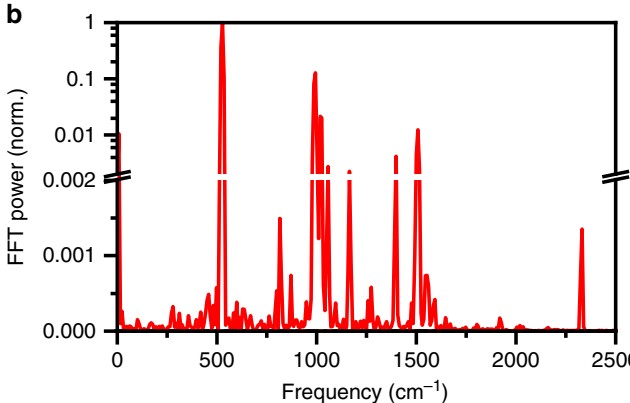

**Fig. 5 First-moment time trace of photodetached aqueous phenoxide. a** First-moment time trace $\langle E^{(1)}(t) \rangle$ obtained following the photodetachment of phenoxide in aqueous solution. **b** Normalized Fast Fourier transform (FFT) power spectrum of the first-moment time trace. Source data are provided as a Source Data file

phase contrasts with the coupling of multiple vibrational modes to the photodetachment of PhO⁻ in aqueous solution. We exclude the possibility that intramolecular structural rearrangement of the PhO· radical is driven by solvent reorganization following photodetachment. Solvent reorganization occurs on timescales that are governed by the frequencies of intermolecular modes, which for the case of liquid water, are in the range of 60–780 cm⁻¹[47]. These intermolecular frequencies are not sufficiently high to drive the impulsive excitation of the phenoxyl radical vibrational modes, the majority of which have frequencies exceeding 800 cm⁻¹.

To rationalize the disparate behavior between gas- and solution-phase results, we employ DFT calculations to determine the extent of structural reorganization upon photodetachment for both the gas phase and solution phase. The results, summarized in Table 2 (see also Supplementary Tables 1–4), indicate that larger structural changes accompany photodetachment in aqueous solution than in the gas phase. For example, while photodetachment in the gas phase leads to a contraction of the C–O bond length by 1.6 pm, the same bond decreases by 5.8 pm upon photodetachment in aqueous solution. The particularly pronounced change in the C–O bond length is most likely due to the presence of water molecules that are hydrogen-bonded to the oxide moiety. Note that similarly large structural changes are observed even when the microhydrating water molecules are frozen in the same geometry upon photodetachment (see Supplementary Tables 5 and 6). This result further excludes the possibility that intramolecular structural rearrangement is driven by solvent reorganization. The overall extent of structural change induced by photodetachment can be quantified by computing the root-mean-square deviation (RMSD), given as

$$\text{RMSD} = \sqrt{\frac{\sum_i^N \left| \boldsymbol{x}_{\text{radical},i} - \boldsymbol{x}_{\text{anion},i} \right|^2}{N}}, \qquad (2)$$

where $\boldsymbol{x}_{\text{anion},i}$ and $\boldsymbol{x}_{\text{radical},i}$ denote the coordinates of the $i$-th atom

**Table 2 Calculated bond lengths of the phenoxide ion and the phenoxyl radical in both gas phase and aqueous solution**

| Bond | Bond length/pm | | Diff./pm | Bond length/pm | | Diff./pm |
|---|---|---|---|---|---|---|
| | PhO⁻ (g) | PhO• (g) | | PhO⁻ (aq) | PhO• (aq) | |
| C1–C2 | 144.7 | 145.2 | −0.5 | 141.7 | 144.8 | −3.1 |
| C2–C3 | 138.8 | 137.5 | 1.3 | 139.3 | 137.2 | 2.1 |
| C3–C4 | 140.3 | 140.8 | −0.5 | 139.8 | 141.1 | −1.3 |
| C4–C5 | 140.3 | 140.8 | −0.5 | 139.8 | 141.1 | −1.3 |
| C5–C6 | 138.8 | 137.5 | 1.3 | 139.3 | 137.2 | 2.1 |
| C6–C1 | 144.7 | 145.2 | −0.5 | 141.7 | 144.8 | −3.1 |
| C1–O7 | 126.9 | 125.3 | 1.6 | 132.6 | 126.8 | 5.8 |

Source data are provided as a Source Data file.

in the $PhO^-$ anion and in the $PhO\cdot$ radical, respectively, and $N$ is the total number of atoms ($N = 12$ for the case of phenoxide). The computed RMSD values – 3.2 pm for gas phase vs. 14.8 pm for aqueous solution—are consistent with structural rearrangement involving more vibrational modes in aqueous solution than in the gas phase.

In conclusion, femtosecond strong-field photodetachment of phenoxide in aqueous solution launches coherent vibrational wave packet dynamics that are resolved into 12 different vibrational frequencies of the phenoxyl radical. Eight of the frequencies are assigned to totally symmetric vibrational modes, along which structural rearrangement occurs, whereas the remaining four vibrational frequencies are assigned to non-totally symmetric modes. The observed coupling of multiple vibrational modes to photodetachment contrasts with the gas-phase photodetachment spectrum of phenoxide, which reveals only a single Franck-Condon-active vibrational mode. The larger extent of photodetachment-induced structural reorganization in aqueous solution is supported by ab initio simulations, which suggest that the RMSD deviation between the $PhO\cdot$ radical and the $PhO^-$ anion geometries is ~5× larger in aqueous solution than in the gas phase. This result points to the important role that the solvent environment plays in determining structural reorganization upon photoionization or photodetachment. The vibrational wave packet dynamics observed in the present work provide, to the best of our knowledge, a first glimpse into the ultrafast structural rearrangement dynamics that accompanies the interaction of ionizing radiation with a molecule in aqueous solution. This time-domain approach can, in principle, be easily extended to larger molecules, including biomolecules, as well as core-ionized species.

## Methods

**Experimental**. Intense, few-cycle laser pulses in the visible are employed to photodetach aqueous phenoxide via a strong-field multiphoton process. Strong-field photodetachment confines the production of phenoxyl radicals to the sub-half-cycle timescale (1.1 fs for the present experimental conditions), hence ensuring a vertical detachment process. These few-cycle pulses are produced by self-phase modulation of the 4-mJ, 1-kHz, 30-fs output from a Ti:sapphire amplified laser system (Coherent, Legend Elite Duo-USX) in a helium-filled hollow-core fiber followed by chirped mirror compression (Ultrafast Innovations GmbH, PC1332). 20% of the compressed output further passes through a $4f$ pulse shaper equipped with a 128-element spatial light modulator (Biophotonics Solutions, FemtoJock-P), which truncates the wavelength range of the chirped mirror compressor output to 530–740 nm, with a central wavelength of 637 nm (1.95 eV photon energy; see Supplementary Fig. 3). This pulse shaper furnishes transform-limited, 6-fs pulses at the sample target (see Supplementary Fig. 4). The maximum pulse energy available for experiments is 25 µJ. A spherical mirror of 50-cm focal length focuses these pulses to a beam waist of 80 µm, resulting in a peak intensity of $4 \times 10^{13}$ W/cm² for the photodetachment pump pulse. An achromatic half-wave plate (B. Halle, RAC4.2.15 L) positioned after the spherical mirror permits the acquisition of differential absorption signals with parallel ($\Delta A_\parallel$) and perpendicular ($\Delta A_\perp$) relative polarization between pump and probe beams. These signals are then used to reconstruct the isotropic signal via the relation $\Delta A = (\Delta A_\parallel + 2\Delta A_\perp)/3$; all the data

reported herein are based on the isotropic $\Delta A$. An optical delay line that is driven by a piezo translation stage (Physik Instrumente, N-664.3 A) is positioned in the pump beam path to provide a variable time delay between the pump and probe pulses in 5-fs steps.

The remaining output from the chirped mirror compressor passes through a pair of wedges (Femtolasers, OA124), followed by a 50-µm-thick Type I beta-barium borate (BBO) crystal (Casix, $\theta = 29.0$ deg) to generate the near-UV probe beam via second harmonic generation. The wedge insertion is adjusted to obtain a broadband near-UV spectrum spanning 360–450 nm (Supplementary Fig. 3). The residual fundamental beam is eliminated by a pair of dichroic beamsplitters (Layertec, 109433) before the near-UV beam is split by a reflective neutral density filter (Newport, FRQ-ND05) into the probe and reference beams. An off-axis parabolic reflector of 15-cm focal length focuses the probe beam to a beam waist of 26 µm at the sample target, after which it is recollimated before it is focused into a 300-mm spectrograph (Princeton Instruments, Acton SP-300) equipped with a 600 grooves/mm grating and a 1024-element linear array detector (Stresing, S8381-1024Q). The spectral resolution in the wavelength range of interest is $\Delta\lambda = 0.3$ nm. The read-out of the array detector is 1 kHz and is synchronized to the 0.5-kHz optical chopper positioned in the path of the ionizing pump beam. The probe pulse energy, measured just before the sample target, is 100 nJ. An identical combination of spectrograph and array detector is used to record the spectrum of the reference beam at 1 kHz repetition rate. This single-shot referencing approach suppresses noise caused by shot-to-shot fluctuations of the probe beam. The focal spot size of the probe beam is ~3× smaller than that of the pump beam, hence minimizing the effect of spatial averaging on the observed ultrafast dynamics. The cross-correlation of the pump and probe beams in a 10-µm-thick Type I BBO crystal (Castech, $\theta = 29.2°$) yields an instrument response of 12 fs at full-width at half-maximum (Supplementary Fig. 4).

The sample target comprises a 7-µm-thick microjet (Metaheuristics, Type L) of aqueous sodium phenoxide operated in air under ambient conditions. The use of a thin microjet limits the group-velocity mismatch between the pump and probe beams to 0.8 fs. Moreover, the vertical flow rate of the microjet (8.5 mm/ms) is sufficiently high to ensure that each laser pulse encounters a fresh focal volume. Sodium phenoxide (Sigma Aldrich, 99%) was used as received. Aqueous solutions of 0.2-M concentration were prepared by dissolving sodium phenoxide in distilled $H_2O$. Fluence dependence measurements suggest that photodetachment proceeds via a 4-photon process (Supplementary Fig. 1), corresponding to a total energy deposition of 7.8 eV, above the threshold of 6.1 eV that is needed to remove an electron from phenoxide (vertical ionization potential 7.1 eV)[23] and inject it into the conduction band of liquid water (electron affinity 0.97 eV)[48].

**Theoretical**. Density functional theory (DFT) calculations employ the GAUSSIAN 09 package[49]. Gradient corrections are introduced in a self-consistent manner by using the three-parameter hybrid exchange functional of Becke (B3)[50] and the correlation functional of Lee, Yang, and Parr (LYP)[51]. The split-valence $6-311G + + (d,p)$ basis set is used. Aside from the isolated phenoxide ion and phenoxyl radical, DFT calculations are also performed on the micro-hydrated species, in which three water molecules are hydrogen bonded to the phenoxide/phenoxyl oxygen atom. To mimic the dielectric environment of the water solvent, the microhydrated species are further embedded in a polarizable continuum[52]. The water molecules are fully relaxed during the geometry optimization of the microhydrated PhO⁻ and PhO• species. Vibrational frequency calculations carried out on the optimized geometries confirm that the structures correspond to local minima. The optimized geometries are given in Supplementary Tables 1–4.

## Data availability

The source data underlying Figs. 3–5, Supplementary Figs. 1–6, and Supplementary Tables 1–6 are provided as a Source Data file.

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

## Acknowledgements

This work was supported by a NTU start-up grant, the Ministry of Education Academic Research Fund (MOE2014-T2-2-052, MOE2018-T2-1-081, and RG105/17), and the award of a Nanyang Assistant Professorship to Z.-H.L.

## Author contributions

T.D. performed the experiment. M.S.B.M.Y. performed the theoretical calculations. T.D., P.J.L., and Z.-H.L. designed and constructed the experimental setup. T.D. and Z.-H.L. wrote the manuscript, with input from all the authors.

## Additional information

**Competing interests:** The authors declare no competing interests.

