## [Peer Review File · Nature Communications]

Reviewers' comments:

Reviewer #1 (Remarks to the Author):

This paper outlines the photodetachment from the phenolate anion in an aqueous solution and the subsequent probing of the vibrational dynamics of the resultant phenoxy-radical. The experiment is rather impressive in terms of its time-resolution and the fact that such clear vibrational wavepacket dynamics are seen. The authors compare the Fourier transform of the wavepacket to the gas-phase photoelectron spectrum and discuss the differences between the two. Overall, this is a very nice study that may be of interest to a wide community interested in solution-phase photo-chemistry. Overall, I find the work quite convincing and the analysis appears sound, but some aspects of the results have left me a little confused.

1) I have looked at the gas-phase spectrum and the signal to noise is this would not be able to distinguish peaks of intensities less than 1% of the main $\nu(11)$ mode. As essentially all of the other modes discussed in the present paper correspond to vibrations with $<1\%$ intensity, I am not convinced that definitively stating that they are not seen in the gas-phase is appropriate. (indeed weaker features are seen but not assigned)

2) the gas-phase spectrum also has a progression from the $\nu(14)$ mode which is of a_2 symmetry. This is not commented on and indeed not seen in the solution phase. Could the authors comment why?

3) my main criticism is the fact that this experiment uses several photons to drive the ionisation and the consequences of this. There is a bright pp^* state that would be accessed with 3 photons. Electron emission from this state has been probed before (Ref 13 in manuscript and a very recent study in JPCB DOI:10.1021/acs.jpcc.8b11766). Both these studies show that emission from this excited state is a charge-transfer process and some time is required to form the radical. Indeed, the present study also shows an increasing signal. It peaks around 1 ps. If the detachment was instantaneous, why would the yield of radical increase over the first ps? To me, this suggests that the pp^* is involved. How certain are the authors that this is not the case (other than their power dependence study)? And what would be the consequences?

4) given that the hydrated electron is also formed, will this not have an impact on the radical dynamics? It would if it was solvated as a contact-pair.

5) I was really surprised to see how long the coherence survived in water (and how narrow the $\nu(11)$ band is in the FFT. This implies that, over the >4 ps where the wavepacket remains coherent, there is almost no structural rearrangement of the water (which one might expect would change the vibrational frequencies) and no IVR.

6) Related to point 5), the $\nu(11)$ mode appears to be very (very) harmonic. According to the gas-phase spectrum, one should be able to see the anharmonicity in this mode and thus a peak at lower frequency to the 528 cm^{-1} band, arising from the $\nu=1-2$ beat.

Reviewer #2 (Remarks to the Author):

This paper describes what is, to the best of my knowledge, a new pump-probe methodology to observe the vibrational dynamics of an ionised molecule or ion (this case) following photoionisation/detachment. I believe it is of sufficient novelty to be suited to Nature Comms. The experimental approach is explained well enough for a general science journal.

There are a few points that I think should be addressed by the authors.

1. In the 1st paragraph, the authors make the comment that "the ultrafast vibrational dynamics of the molecular radical species remains largely unknown" - since this is a paper for a general audience, I think it would be worth explaining here why this it is important to know what happens, perhaps giving some examples relevant to biological systems. The next sentence about x-ray emission spectra is interesting but how relevant is it that removal of an electron from an O $1s$ core hole induces proton transfer to what happens after a valence electron is removed? I think its worth keeping this in the introduction but the link needs to be made better.

2. My most important comment is really about what the vibrational frequencies that are observed in these experiments actually mean. On p3, the authors state that in the solution phase measurement the vibrational frequencies that appear in the pumo-probe signal encode the identity of normal modes along which structural rearrangement occurs upon photodetachment. I have two queries about this statement, that the rest of the paper is based on.

(i) How sure are the authors that the multi photon process they employ to photodetach phenoxide does not involve resonances with excited states of the phenoxide anion? There is a high density of

excited states lying in the detachment continuum of phenoxide in the gas-phase (see e.g. DOI: [acs.jpca.8b11121](https://doi.org/10.1021/acs.jpca.8b11121)) and in aqueous solution, these will all be below the detachment threshold and easily accessed in this experiment at the 2 or 3 photon level.

(ii) Is it the structural rearrangement that occurs upon photodetachment that is being observed or is it the structural rearrangement that occurs after photodetachment due to relaxation of the solvent? I didn't quite follow what the calculation was that the authors did for the detachment process? Did they relax the water molecules around the phenoxide radical? Or did they do the calculation with the water molecules in the same place as for the anion? It would be interesting to compare the two different calculations so then it would be possible to determine if the 12 vibrational modes activated by detachment in solution (compared to predominantly 1 in the gas-phase) is a result of the phenoxide being H-bonded to water molecules (as stated by the authors) or the result of the subsequent relaxation of the water molecules after the detachment process.

If these comments are addressed I am confident the paper would be suitable for publication.

Reviewer #1

This paper outlines the photodetachment from the phenolate anion in an aqueous solution and the subsequent probing of the vibrational dynamics of the resultant phenoxy-radical. The experiment is rather impressive in terms of its time-resolution and the fact that such clear vibrational wavepacket dynamics are seen. The authors compare the Fourier transform of the wavepacket to the gas-phase photoelectron spectrum and discuss the differences between the two. Overall, this is a very nice study that may be of interest to a wide community interested in solution-phase photo-chemistry. Overall, I find the work quite convincing and the analysis appears sound, but some aspects of the results have left me a little confused.

We thank the Reviewer for appreciating our work and for the thoughtful comments. We hope that our response herein, along with the accompanying revised manuscript, would help to clarify the Reviewer's doubts.

1) I have looked at the gas-phase spectrum and the signal to noise is this would not be able to distinguish peaks of intensities less than 1% of the main $\nu(11)$ mode. As essentially all of the other modes discussed in the present paper correspond to vibrations with <1% intensity, I am not convinced that definitively stating that they are not seen in the gas-phase is appropriate. (indeed weaker features are seen but not assigned)

We thank the Reviewer for this important comment. We agree with the Reviewer that it would be difficult to identify peaks with amplitudes that are <1% that of the main ν_{11} peak (519 cm^{-1}) in the gas-phase spectrum (ref. 27), and therefore we should not state that only a single vibrational mode was observed in the gas-phase measurement. We have clarified this point on p. 12 of the revised manuscript (REV 1.1).

2) the gas-phase spectrum also has a progression from the $\nu(14)$ mode which is of a2 symmetry. This is not commented on and indeed not seen in the solution phase. Could the authors comment why?

We thank the Reviewer for raising this question. Indeed, the Franck-Condon progression in the gas-phase study (ref. 27) comprises two series of peaks: one for 11_0^m and another one for $14_1^1 11_0^m$. The latter corresponds to a series of sequence bands involving the ν_{14} mode trailing the ν_{11} progression. These bands are present because the vibrational temperature of the phenoxide precursor, estimated to be $\sim 300\text{ K}$, gives rise to a finite population of the ν_{14} mode ($\nu_{14} = 420\text{ cm}^{-1}$) in the phenoxide precursor. Please note that there is no change in the vibrational quantum number of the ν_{14} mode accompanying photodetachment.

Peaks that appear in the photodetachment spectra, after subtracting the electron affinity offset, correspond to differences between the vibrational energy levels of the phenoxide anion and the phenoxy radical. This explains the -51 cm^{-1} downshift in the position of the sequence bands relative to the ν_{11} progression. In our measurements by femtosecond wave packet spectroscopy, the oscillation frequencies that are observed correspond to differences between the vibrational energy levels of the phenoxy radical itself, i.e., the wave packet motion involves only the phenoxy radical. As such, we do not expect to observe vibrational

frequencies that are associated with the sequence bands that are identified in the gas-phase study. We have added this clarification to the Supplementary Information as Supplementary Note 3 (REV 1.2).

3) my main criticism is the fact that this experiment uses several photons to drive the ionisation and the consequences of this. There is a bright pp^* state that would be accessed with 3 photons. Electron emission from this state has been probed before (Ref 13 in manuscript and a very recent study in JPCB DOI:10.1021/acs.jpcb.8b11766). Both these studies show that emission from this excited state is a charge-transfer process and some time is required to form the radical. Indeed, the present study also shows an increasing signal. It peaks around 1 ps. If the detachment was instantaneous, why would the yield of radical increase over the first ps? To me, this suggests that the pp^* is involved. How certain are the authors that this is not the case (other than their power dependence study)? And what would be the consequences?

We thank the Reviewer for this important comment. As alluded to by the Reviewer, our pump-power dependence study shows that photodetachment proceeds via a four-photon process. If the phenoxyl radical was produced via the S_1 ($\pi\pi^*$) state, the power dependence measurement would have revealed a three-photon process instead. Such a three-photon process was not observed. Moreover, contributions from excited-state dynamics in the presence of an intense laser field should be negligible because these states are located in close energy proximity to the photodetachment threshold and should therefore undergo rapid strong-field-induced photodetachment. Given the 6-fs pulses that are employed in strong-field photodetachment, we do not expect any excited states of phenoxide to survive beyond the 6-fs duration of the intense pump pulse. We have clarified this on p. 6 of the revised manuscript and inserted two accompanying references (refs. 34 & 35) (REV 1.3A).

We also thank the Reviewer for bringing to our attention the recent publication by the Verlet group (DOI:10.1021/acs.jpcb.8b11766). We note that previous studies have shown that photoexcitation to the S_1 state of phenoxide furnishes the phenoxyl radical. In ref. 24, photodetachment of phenoxide with 266-nm light furnished time constants of 5 ps and 37 ps, which were assigned to electron ejection from the hot and relaxed S_1 state, respectively. In the work by Verlet, photodetachment with 257-nm light yielded a differential absorption signal that grew in with a time constant of 0.58 ps. On p. 5 of the revised manuscript, we briefly discuss the appearance of our time-resolved differential absorption spectrum (Figure 2a) in the context of these earlier results; we also insert the article by the Verlet group as a new reference (ref. 32) (REV 1.3B).

Regarding the Reviewer's comment about the rising differential absorption signal that is observed in our data, we would like to point out that the same increase is also observed in our control experiments performed on pure liquid water under experimental conditions identical to those employed for aqueous phenoxide. Please see Figure R1 below. At the probe wavelength of 406 nm, for example, the time constants for pure liquid water and aqueous phenoxide are 0.57 ± 0.01 ps and 0.54 ± 0.01 ps, respectively, almost identical to within experimental error. It is known that the rising differential absorption observed in ionized pure liquid water originates from the formation of the hydrated electron (see refs. 36 & 37).

Figure R1 | Comparison of the time-resolved differential absorption signals of water and aqueous phenoxide. The differential absorption time traces collected at 406 nm for water (blue line) and aqueous phenoxide (red line) both show a rise that can be fit to time constants of 0.57 ± 0.01 ps and 0.54 ± 0.01 ps, respectively. Also note that the time traces, when extrapolated to 0 fs, reveal an offset for aqueous phenoxide, indicating the instantaneous formation of the phenoxyl radical by strong-field photodetachment of phenoxide.

Because the hydrated electron has a finite formation time (~ 0.5 ps) and exhibits a broad absorption feature spanning 300 – 1000 nm, the photodetachment of phenoxide, which also yields a hydrated electron, should result in a rising absorption signal across the entire probe spectral range employed in our study. Given the matching rise times observed in pure liquid water and aqueous phenoxide, we believe that rising absorption signal is not due to delayed formation of the phenoxyl radical via the S_1 state of phenoxide, but rather, originates from hydrated electron formation. Further evidence for the immediate production of the phenoxyl radical via strong-field photodetachment comes from the time traces in Figure 2b, which show an offset immediately following time-zero. This offset, absent in the time traces collected for ionized pure liquid water, is consistent with the instantaneous formation of the phenoxyl radical by strong-field photodetachment. We have inserted the above information into p. 6 – 7 of the revised manuscript (REV 1.3C). Figure R1 has also been added to the Supplementary Information as Supplementary Figure 2.

4) given that the hydrated electron is also formed, will this not have an impact on the radical dynamics? It would if it was solvated as a contact-pair.

The formation of the hydrated electron on the 0.5-ps timescale occurs significantly after the vibrational wave packet dynamics of the phenoxyl radical have been initiated. As such, we do not expect contact pair formation to affect the launching of the vibrational wave packet dynamics.

5) I was really surprised to see how long the coherence survived in water (and how narrow the $\nu(11)$ band is in the FFT. This implies that, over the >4 ps where the wavepacket remains coherent, there is almost no structural rearrangement of the water (which one might expect would change the vibrational frequencies) and no IVR.

We thank the Reviewer for this comment. Indeed, our time-domain analysis of the first-moment time trace (Figure 4a) suggests that the ν_{11} oscillation has a dephasing time of $1.86 \pm$

0.07 ps. All the other observed vibrational modes have dephasing times exceeding >0.5 ps. As the Reviewer suggests, these long dephasing times are suggestive of slow intramolecular vibrational relaxation (IVR) and a negligible effect of the solvent rearrangement on the vibrational frequencies of the phenoxyl radical. We note that similarly long dephasing times have been observed for other molecules in the condensed phase. For example, broadband optical pump-probe measurements of cresyl violet in methanol yielded a dephasing time of 2.4 ps for the vibrational mode that dominates the wave packet signal [*J. Phys. Chem. A* **120**, 6792–6799 (2016)]. Similar experiments on methylene blue in aqueous solution observed clear vibrational coherences beyond the 2-ps time window of the experiment [*J. Phys. Chem. A* **120**, 9098–9108 (2015)]. We have inserted the above information and the accompanying references into p. 10 of the revised manuscript (REV 1.5).

6) Related to point 5), the $\nu(11)$ mode appears to be very (very) harmonic. According to the gas-phase spectrum, one should be able to see the anharmonicity in this mode and thus a peak at lower frequency to the 528 cm^{-1} band, arising from the $\nu=1-2$ beat.

We thank the Reviewer for this comment. According to the gas-phase photodetachment study (ref. 27), the first overtone of the ν_{11} mode is located at 1038 cm^{-1} , exactly double that of the fundamental frequency at 519 cm^{-1} . This result suggests that anharmonicity of the ν_{11} mode is very small. Our assignment of the 1054- cm^{-1} mode to the first overtone of the 528- cm^{-1} mode is consistent with the vanishing anharmonicity for this mode. In fact, the small anharmonicity suppresses coupling between different vibrational modes, thereby supporting slow IVR, and hence, slow dephasing of the vibrational wave packets. We have added this clarification to p. 11 – 12 of the revised manuscript (REV 1.6).

Reviewer #2

This paper describes what is, to the best of my knowledge, a new pump-probe methodology to observe the vibrational dynamics of an ionised molecule or ion (this case) following photoionisation/detachment. I believe it is of sufficient novelty to be suited to Nature Comms. The experimental approach is explained well enough for a general science journal.

We are grateful to the Reviewer for supporting the publication of this work in *Nature Communications*. We hope that our response below addresses the concerns of the Reviewer.

There are a few points that I think should be addressed by the authors.

1. In the 1st paragraph, the authors make the comment that "the ultrafast vibrational dynamics of the molecular radical species remains largely unknown" - since this is a paper for a general audience, I think it would be worth explaining here why this it is important to know what happens, perhaps giving some examples relevant to biological systems. The next sentence about x-ray emission spectra is interesting but how relevant is it that removal of an electron from an O 1s core hole induces proton transfer to what happens after a valence electron is removed? I think its worth keeping this in the introduction but the link needs to be made

better.

We thank the Reviewer for this valuable suggestion. We have expanded the introductory paragraph to mention that ultrafast structural rearrangement impacts the chemical reactivities, redox potentials, and electron transfer rates involving the radical species, citing several studies of biological systems involving long-range electron transfer in redox proteins and charge transport in DNA (REV 2.1A).

Furthermore, please allow us to clarify that, in our opinion, it is important to study the ultrafast vibrational dynamics induced by both valence- and core-level ionization. Hence, even though we have focused on valence photodetachment in the present work, our approach of using femtosecond vibrational coherence spectroscopy to probe ultrafast structural rearrangement can, in principle, be applied to core-ionized species as well. We now clarify this in the concluding paragraph of the revised manuscript (REV 2.1B).

2. My most important comment is really about what the vibrational frequencies that are observed in these experiments actually mean. On p3, the authors state that in the solution phase measurement the vibrational frequencies that appear in the pump-probe signal encode the identity of normal modes along which structural rearrangement occurs upon photodetachment. I have two queries about this statement, that the rest of the paper is based on.

(i) How sure are the authors that the multi photon process they employ to photodetach phenoxide does not involve resonances with excited states of the phenoxide anion? There is a high density of excited states lying in the detachment continuum of phenoxide in the gas-phase (see e.g. DOI: acs.jpca.8b11121) and in aqueous solution, these will all be below the detachment threshold and easily accessed in this experiment at the 2 or 3 photon level.

We thank the Reviewer for this important comment, which is similar to comment #3 made by Reviewer #1. Please kindly see above for our response to the same comment made by Reviewer #1. In addition, we thank the Reviewer for bringing to our attention the recent work by Fielding and co-workers, which we have inserted as ref. 33 in the revised manuscript (REV 2.2A).

(ii) Is it the structural rearrangement that occurs upon photodetachment that is being observed or is it the structural rearrangement that occurs after photodetachment due to relaxation of the solvent? I didn't quite follow what the calculation was that the authors did for the detachment process? Did they relax the water molecules around the phenoxide radical? Or did they do the calculation with the water molecules in the same place as for the anion? It would be interesting to compare the two different calculations so then it would be possible to determine if the 12 vibrational modes activated by detachment in solution (compared to predominantly 1 in the gas-phase) is a result of the phenoxide being H-bonded to water molecules (as stated by the authors) or the result of the subsequent relaxation of the water molecules after the detachment process.

We thank the Reviewer for this comment. We believe that structural rearrangement is driven by sudden photodetachment and not by solvent reorganization following photodetachment. Solvent reorganization occurs on timescales that are governed by the frequencies of intermolecular modes, which for the case of liquid water, are in the range of 60 – 780 cm^{-1} [*J. Chem. Phys.* **40**, 3249–3256 (1964)]. These intermolecular frequencies are not sufficiently high to drive the impulsive excitation of the phenoxyl radical vibrational modes, the majority of which have frequencies exceeding 800 cm^{-1} . The above is clarified on p. 13 of the revised manuscript (REV 2.2B1).

In the ab initio calculation of the microhydrated species, the three water molecules were relaxed for both phenoxide and phenoxyl radical. We have clarified this in the Theoretical Methods section (REV 2.2B2).

Per the Reviewer's suggestion, we repeated the calculation on the microhydrated phenoxyl radical, with the water molecules and the oxygen atom of the phenoxyl radical frozen in the same geometry as in the case of the microhydrated phenoxide species. The results are summarized in Table R1 below. For ease of comparison, we also show the original calculation results for gas-phase photodetachment.

Table R1. Results of DFT Calculations, Showing the Bond Lengths of the Phenoxide Ion and the Phenoxyl Radical, and the Difference Bond Lengths, in Both Gas Phase and Aqueous Solution.

Bond	Bond Length / pm		Diff. / pm	Bond Length / pm		Diff. / pm
	PhO ⁻ (g)	PhO• (g)		PhO ⁻ (aq)	PhO• (aq)*	
C1 – C2	144.7	145.2	-0.5	141.7	144.6	-2.9
C2 – C3	138.8	137.5	1.3	139.3	137.2	2.1
C3 – C4	140.3	140.8	-0.5	139.8	141.2	-1.4
C4 – C5	140.3	140.8	-0.5	139.8	141.2	-1.4
C5 – C6	138.8	137.5	1.3	139.3	137.1	2.2
C6 – C1	144.7	145.2	-0.5	141.7	144.6	-2.9
C1 – O7	126.9	125.3	1.6	132.6	127.8	4.8

* The microhydration water molecules and H₂O•••O(phenoxide) distances are not relaxed upon photodetachment.

From the Table R1, we can see that the phenoxyl radical undergoes large structural rearrangement even when the microhydrating water molecules are frozen in the same geometry upon photodetachment. This result excludes the possibility that intramolecular structural rearrangement is driven by solvent reorganization. We have included this information on p. 14 of the revised manuscript (REV 2.2B3). Table R1 is added to the Supplementary Information as Supplementary Table 1 and the partially optimized coordinates of the phenoxyl radical with the frozen microhydrating water molecules is added as Supplementary Table 6.

If these comments are addressed I am confident the paper would be suitable for publication.

REVIEWERS' COMMENTS:

Reviewer #1 (Remarks to the Author):

The authors have addressed the vast majority of my comments and have clarified several points. There one point I'd just like to comment on. In point (4), I only partly agree with the response. The reason why the hydrated becomes "visible" in the probe range after 0.5 ps is because it thermalizes on this timescale with a concomitant blue shift of the absorption spectrum. The hydrated electron is formed on a much shorter timescale. I suspect that the electron is emitted at longer range (as is the case in ionization of pure water) and solvates away from the radical so that no contact pair is formed. Perhaps a short statement considering the fate of the electron would be useful. This is not a major point, but it would be useful to clarify.

Overall, this is a very nice contribution and I recommend the manuscript be accepted.

Reviewer #2 (Remarks to the Author):

I have read the revised manuscript carefully, along with the response to the comments of both reviewers and I am convinced that the authors have addressed all the points very clearly and that the manuscript is now very clear. The paper describes work that I am sure will be of interest to a wide range of spectroscopists and theoreticians interested in ionisation and should be published as it is in Nature Communications.

Reviewer #1

The authors have addressed the vast majority of my comments and have clarified several points. There one point I'd just like to comment on. In point (4), I only partly agree with the response. The reason why the hydrated becomes "visible" in the probe range after 0.5 ps is because it thermalizes on this timescale with a concomitant blue shift of the absorption spectrum. The hydrated electron is formed on a much shorter timescale. I suspect that the electron is emitted at longer range (as is the case in ionization of pure water) and solvates away from the radical so that no contact pair is formed. Perhaps a short statement considering the fate of the electron would be useful. This is not a major point, but it would be useful to clarify.

Overall, this is a very nice contribution and I recommend the manuscript be accepted.

We thank the Reviewer for this comment and for recommending the acceptance of the manuscript. We agree that the spatial delocalization of the initially produced electron is unlikely to yield a contact pair with the phenoxyl radical. We have clarified this by inserting the following into p. 5 – 6 of the revised manuscript. “The precursor to the hydrated electron is an electron that is initially injected into the conduction band of liquid water by ionization. Due to its spatial delocalization, the formation of a contact pair between the conduction-band electron and the phenoxyl radical is not expected to occur. As such, the observed ultrafast dynamics is unlikely to originate from such a contact pair.”

Reviewer #2

I have read the revised manuscript carefully, along with the response to the comments of both reviewers and I am convinced that the authors have addressed all the points very clearly and that the manuscript is now very clear. The paper describes work that I am sure will be of interest to a wide range of spectroscopists and theoreticians interested in ionisation and should be published as it is in Nature Communications.

We thank the Reviewer for supporting our work for publication.